# Canyoning Accidents in Austria from 2005 to 2018

**DOI:** 10.3390/ijerph17010102

**Published:** 2019-12-22

**Authors:** Mathias Ströhle, Ina Beeretz, Christopher Rugg, Simon Woyke, Simon Rauch, Peter Paal

**Affiliations:** 1Department of General and Surgical Critical Care Medicine, Medical University of Innsbruck, Anichstrasse 35, 6020 Innsbruck, Austria; christopher.rugg@tirol-kliniken.at; 2Social Services Vienna, Centre for Paediatric Development, 1220 Vienna, Austria; inabeeretz@icloud.com; 3Institute for Mountain Emergency Medicine, EURAC Research, via Ipazia 1, 39100 Bolzano, Italy; simon.Woyke@eurac.edu (S.W.); simon.Rauch@eurac.edu (S.R.); 4Department of Anaesthesiology and Intensive Care Medicine, Hospitallers Brothers Hospital, Paracelsus Medical University Salzburg, Kajetanerplatz 1, 5010 Salzburg, Austria; peter.paal@icloud.com

**Keywords:** canyoning, accidents, emergency medical service, search and rescue, epidemiology, trauma and injuries

## Abstract

Canyoning has become a popular recreational sport. Nevertheless, little is known about injuries or diseases associated with canyoning. The aim of this study was to examine accident causes, injury patterns, out-of-hospital and in-hospital treatment and outcomes. For this purpose, national out-of-hospital data from the Austrian Alpine Safety Board and regional in-hospital data from Innsbruck Medical University Hospital were analysed for the period from November 1, 2005 to October 31, 2018. Nationally, 471 persons were involved in such accidents; 162 (34.4%) were severely injured, nine of whom died. Jumping (*n* = 110, 23.4%), rappelling (*n* = 51, 10.8%), sliding (*n* = 41, 8.7%) and stumbling (*n* = 26, 5.5%) were the most common causes of canyoning accidents. A large proportion of injuries were documented for the lower extremities (*n* = 133, 47.5%), followed by the upper extremities (*n* = 65, 23.2%) and the spine (*n* = 44, 15.7%). Death was mainly caused by drowning. Overall mortality was 1.9% (*n* = 9), and the absolute risk was 0.02 deaths per 1000 hrs of canyoning. Many uninjured persons required evacuation (*n* = 116, 24.6%), which resulted in a substantial expense and workload for emergency medical services. Increased safety precautions are required to reduce accidents while jumping and rappelling and fatalities caused by drowning.

## 1. Introduction

Canyoning is a wilderness recreational sport that has grown in popularity, especially in Europe, the USA and Australia [1,2]. Canyoning requires water and rock expertise, combining diverse skills, such as hiking up and down wet and dry rocks, scrambling, climbing, bouldering and roping techniques like rappelling, as well as swimming, sliding, jumping, or diving off cliffs or waterfalls. Principal risks, such as falls onto rocks, drowning and hypothermia, are obvious, but comprehensive literature on dangers and injury patterns in canyoning is scarce. Typical pathologies, such as the ‘Canyoning-Hand,’ a form of contact dermatitis with skin lesions involving only the fingertips and palms, due to the cold, wet and rugged nature of canyoning, have been described [3]. In rare cases, infectious diseases, e.g., leptospirosis or tularemia, can also be canyoning-related and even occur as a local outbreak [4,5,6]. A study conducted in the Autonomous Community of Aragon, Spain, collected data for a ten-year period in a typical canyoning region [7]. The study reported 80.6% trauma-related injuries vs. 9.4% illnesses and no injuries in 9.6% of the study patients, and highlighted that the administration of analgesics (33.7%) and splinting (71.2%) during rescue were most commonly performed by emergency physicians. Another study in Zion National Park (UT) reported 211 search and rescue missions over a ten-year period [8]. In 51% of all cases, trauma was the reason for search and rescue missions, resulting in 40% of the overall costs for search and rescue missions in Zion National Park. Both studies lack information on the accident causes and do not report the activities being pursued during canyoning that resulted in the injuries.

In 2017, the International Commission for Mountain Emergency Medicine (ICAR MEDCOM) defined consensus guidelines for the on-site management and transport of patients in canyoning accidents, as well as summarising, commenting on and grading the literature available [9]. Although regional data have been reported, no long-term national data on canyoning have been published to date. The aim of this study was to analyse nationwide canyoning data for Austria covering the period from November 1, 2005 to October 31, 2018.

## 2. Materials and Methods 

This retrospective study was approved by the Ethics Committee of the Medical University of Innsbruck (AN4757 315/4.4) and was registered with Clinical Trials (NCT03405467). In Austria, Alpine Police officers collect and store anonymised data on accidents and rescue operations. These officers are members of the Federal Ministry of the Interior, are qualified alpinists and have basic paramedical training. Whenever an emergency call triggers a professional rescue, an entry into the national registry for mountain accidents in Austria is made. By October 31, 2018, this registry contained more than 109,710 mountain emergencies, thus being one of the largest registries for mountain accidents. For the purpose of this study, accidents related to canyoning during the timeframe from initiation of the registry on November 1, 2005 to October 31, 2018, were extracted and analysed. For in-hospital data and patient outcome, the hospital information system of Innsbruck Medical University Hospital, a level-one trauma centre, was analysed. 

In a first step, registry data containing the sex, age, and nationality of involved persons; circumstances and accident cause; body region; type and severity of injury; type of extrication and rescue; weather; location; elevation; and several other data were collected (Appendix A). The epidemiology of all victims involved in canyoning accidents occurring during this thirteen-year period is described. Severity of injury was defined as mild when the patient was unable to work for less than 24 days. A severe injury was defined by Austrian lawmakers as any fracture, except a fracture of the nose, or an inability to work for 24 days or more after the accident [10]. Traumatic brain injury was defined as mild, moderate or severe, according to the American Congress of Rehabilitation Medicine (ACRM) definition [11,12]. Accidental hypothermia was graded in accordance with the Swiss Staging of Hypothermia. Accidental hypothermia I is defined as a body temperature of 32 to 35 °C [13]. Lesions were any injuries to the tissue not defined as any other injury.

In a second step, the local hospital information system at Innsbruck Medical University Hospital, a level-one trauma centre for Western Austria, was analysed with a full text query for „canyon“, „canyoning” and „canyoning accidents” for the same period (November 1, 2005 to October 31, 2018). Records of patients involved in canyoning accidents, irrespective of their age, were included in this study. Medical information was obtained by searching in-hospital medical reports, emergency physician protocols and investigations like X-rays or CT scans. Persons only swimming in a gorge or not doing canyoning were not included. Hospital information systems at other Austrian hospitals were not analysed due to restrictions imposed by the Austrian Data Protection Authority. 

Steps involved for data acquisition from the national registry of the Austrian Alpine Safety Board and from the hospital information system at Innsbruck Medical University Hospital are depicted in a CONSORT Flowchart (Appendix A). Data are presented as the mean ± SD or absolute and relative frequencies, as appropriate. Both national registry and Innsbruck Medical University Hospital data are analysed and presented separately. Data from both queries were collected with Excel 2019 (Microsoft, Seattle, WA) and analysed with SPSS 24.0 (IBM, Armonk, NY, USA).

## 3. Results

### 3.1. National Registry Data from the Austrian Alpine Safety Board

#### 3.1.1. Epidemiology

The registry of the Austrian Alpine Safety Board shows 297 canyoning accidents as accidents, environmental or medical emergencies. Males (73.9%) more often suffered a canyoning accident than females (25.7%). Of the 471 persons involved, 392 (83.2%) were located in Austria’s western states (the catchment area of the Innsbruck Medical University Hospital). Of them, 314 (66.7%) were in the state of Tyrol. Most incidents occurred during the summer (June *n* = 69, 14.6%; July *n* = 131, 27.8%; August *n* = 161, 34.2%; September *n* = 57, 12.1%) and most frequently on Saturdays (*n* = 131, 27.8%) or Sundays (*n* = 90, 19.1%). The median time of day of the emergency was 13:43 ± 2:35 h. Of the study patients, 303 (64.3%) were on a guided canyoning tour. Table 1 reports data on the sex, age, elevation of the accident site, activity before accident, severity of injury, rescue mode and weather.

#### 3.1.2. Pattern and Severity of Injury

In 297 accidents, injuries were suffered by 302 (64.1%) persons, while 169 (35.9%) persons were involved, but not injured. Table 2 summarises the canyoning activities leading to injuries. The risk for suffering severe injuries was greatest for jumps, slides and descents. In 16 cases, the involved persons fell while rappelling or descending from a mean height of 11 ± 12 m. Five individuals fell because of incorrect rope handling. One person fell when an anchor malfunctioned. The maximum height of falls was 38 m while rappelling and 40 m while descending. Where falling was the cause of the accident, all individuals were injured either mildly or severely, and no patient died. The majority of fatal events occurred while swimming. In two cases, the victims remained attached to a rope after rappelling, whilst the other three victims drowned in white-water. One person suffered a cardiac arrest, probably due to exhaustion. Fifty-three people were trapped in a canyon (11.3%). The main injuries were fractures (*n* = 117, 24.8%), strains or sprains (*n* = 79, 16.8%) and dislocations (*n* = 48, 10.2%). The region affected most by injury was the lower extremity (*n* = 133, 47.5%), followed by the shoulder (*n* = 46, 16.4%) and the spine (*n* = 44, 15.7%) From the 302 injured, data on the injured body region were missing in 22 cases. These were excluded, resulting in a total of 280 injured individuals (Table 3 and Table 4).

#### 3.1.3. Mode of Rescue

Injuries to the lower extremity, shoulder and spine were common and frequently prompted rescues by a helicopter. Some persons were uninjured, but needed to be rescued after their companions had been involved in an accident (*n* = 96, 20.4%). Overall, 116 (24.6%) canyoneers, although uninjured, needed to be rescued. Seventy-one persons practicing canyoning were involved in, but not in need of, a rescue mission. Helicopters were frequently dispatched in sunny (*n* = 138, 50.9%) and cloudy (*n* = 55, 61.1%) weather conditions, whereas ground-based rescue missions were more common in bad weather conditions (*n* = 46, 68.6%). The type of rescue in relation to the injured body region is illustrated in Table 4.

### 3.2. Innsbruck Medical University Hospital Data

#### 3.2.1. Epidemiology

Fifty-eight patients who suffered injuries or became medically ill while canyoning were treated at Innsbruck Medical University Hospital. Males (56.9%) were more frequently involved than females (43.1%). Details of the canyoning activity performed when the accident occurred and the required rescue are shown in Table 5.

#### 3.2.2. Injury Pattern and Severity

Table 6 summarises the diagnoses of patients treated at Innsbruck Medical University Hospital. The lower extremity was the most frequently affected anatomical region (*n* = 25, 45.5%), followed by the upper extremity (*n* = 14, 25.5%) and the vertebral column (*n* = 14, 25.5%). Seven surgeries and one percutaneous transluminal coronary angioplasty (PTCA) were necessary (Table 6). Six cases called for surgical debridement, in addition to antibiotics, when lesions occurring in canyoning accidents became infected. Patients with vertebral column fractures suffered no neurological deficit. No surgeries were performed for fractures, sprains or dislocations of the lower extremity. Two cases of gastroenteritis occurred, but no specimens or stool samples were collected.

#### 3.2.3. Mode of Rescue

Only nine (15.5%) patients required a rescue mission of any kind. The other patients (*n* = 48, 82.8%) sought out the emergency department on their own.

## 4. Discussion

By analysing the national registry of the Austrian Alpine Safety Board on alpine accidents, we were able to discern the epidemiology, injury severity and pattern, mortality and mode of rescue of persons involved in canyoning accidents in Austria. Jumping, rappelling, sliding and stumbling were the most common causes of canyoning accidents. Injuries most commonly involved the lower extremity, followed by the shoulder and the spine. Death was mainly caused by drowning. Nearly half of the rescue operations required a helicopter to be dispatched. A quarter of all canyoneers (*n* = 116, 24.6%) were uninjured but needed rescuing, which caused a substantial expense and workload for emergency medical services.

### 4.1. Epidemiology

Of a total of 109,710 registered accidents in the Austrian Alps, 297 were able to be linked to canyoning accidents and involved 471 people. The male to female ratio was 2.9:1, while a study from Zion National Park (UT) reported a ratio of 2.7:1 and a Spanish study from Aragon reported a ratio of 1.4:1 [7,8]. The mean age was 32.8 yrs., which was comparable to other canyoning registries [7,8]. The observed seasonality from June to September and the major focus on weekends show that canyoning in Austria is a seasonal sport. The mainly sunny or cloudy weather at the time of the emergencies (78.8%) shows that bad weather is not a major cause of accidents. Most accidents occurred in the early afternoon; this may be due to operator fatigue during long canyoning adventures. A Swiss online survey conducted by Ernstbrunner et al. showed a correlation between level of experience and occurrence of injuries. Canyoning with professional guides was shown to reduce injury incidence. It should be noted that in the Swiss survey, an injury was defined as any physical complaint, regardless of medical treatment required, which is in contrast to the definition used in our study. Moreover, the Swiss survey was conducted worldwide. The level of canyoneer experience reported in the Swiss study was 29% professionals able to perform independent, non-guided tours, and 70% (28% beginners, 42% advanced) needing guidance, at least on tours requiring climbing and rappelling [14]. The distribution of canyoning experience in the Austrian Alps is unknown. We can only highlight that in our study, two-thirds of all patients were on a guided tour. It seems reasonable that people needing guidance are less experienced than people canyoning on their own, but at the same time, the company of a professional guide would suggest increased safety and reduced operator and planning errors.

Of the 58 patients treated at Innsbruck Medical University Hospital, only nine required a professional rescue (1:6.4). This means that of the 58 persons injured while canyoning, only nine were registered nationally. By extrapolating this ratio to our data from the national registry, the actual total number injured during canyoning can be hypothesised to be 6.4 times higher than those injured and registered nationally (as in requiring a rescue mission). By subtracting the uninjured from the total persons involved in a canyoning emergency nationally, one can calculate those injured and registered (*n* = 302). Therefore, a total of 1933 individuals can be estimated as injured during canyoning in Austria. We assume that the Austrian Alpine Safety Board underestimates the number of patients with minor or no injuries as they can mostly rescue themselves, without alerting the dispatch centre and triggering a professional rescue.

### 4.2. Injury Pattern and Severity

The main activities leading to injury were jumping, rappelling, sliding or descending, as well as stumbling or falling. The prospective survey from Switzerland observed that injuries most commonly occur while rappelling and mainly involved the hand (23%), lower leg or foot (25%) in a nine-month period. They were mostly minor and limited to soft tissues [14]. Another web-based survey of canyoneers described mostly minor cutaneous and locomotor system injuries. Only two of the 38 surveyed persons had required outside assistance during canyoning at any time in their lifetime [1]. Both canyoning surveys also reported material failure being causative in almost half of the cases of documented injury [1,14]. Although our data were collected by police officers who looked for evidence for potential investigations, prosecutions or judicial proceedings, we were able to identify only five cases of incorrect rope handling and one case of anchor malfunction. Injured canyoneers seem to see responsibility in material failure, whereas police officers report operator errors to be more common in their investigations.

With regard to the injury pattern and severity, as noted in the national registry, we observed a large number of severe injuries, a smaller proportion of mild injuries and a minor number of deaths. Cause of death was mainly drowning. The most affected body region was the lower extremity, followed by the shoulder and spine, with the main type of injury being fractures, followed by sprains or strains and dislocations. Similar to the Swiss prospective survey, a retrospective analysis from Aragon, Spain, described 90% of injuries as minor to moderate (NACA 1−3, Appendix A, [15]). Only a total of 6% of the persons studied in Spain were severely injured and 4% died [7]. Notably, our definition of injury severity differed.

Strictly referring to search and rescue operations, a study from Zion National Park (UT) reported a large percentage of uninjured or non-ill (79%) persons [8], while the Spanish study reported merely 10% [7]. We observed more than a quarter of all canyoneers to be uninjured but in need of rescuing. The main reasons for this divergence may be the fact that the different geographic regions involved can be more or less prone to getting trapped or lost and therefore requiring an uninjured rescue, but the registries analysed also differ, such as in the type of rescue and their trigger for dispatch (search and rescue mission vs. rescue operation). This is also reflected in the above-mentioned difference in the proportion of trauma-related injuries between the studies.

The data from Innsbruck Medical University Hospital show that injury primarily occurred to the lower extremity, the upper extremity and the vertebral column. Although other studies have also shown that lower extremity and spinal injuries are common in canyoning, our in-hospital data show a larger proportion of spinal injuries, as well as higher rates of fracture and dislocation in comparison to data from other studies [7,8].

Medical emergencies were few. Gastrointestinal infections were documented in our in-hospital data, but did not permit direct linkage to canyoning. As no microbiological samples were collected in our patients with gastrointestinal infections, it is unknown whether a specific pathogen was causative. Canyoning-related waterborne, infectious diseases, e.g., tularaemia and leptospirosis, have been described in warmer countries [3,6,16,17] and general practitioners, as well as hospital staff, should keep them in mind when treating patients who have practiced water-related activities [4,5]. Preventive measures, such as clothing for skin protection, are an option to prevent infectious diseases [17]. Ill patients should be treated with doxycycline when suspecting leptospirosis [16]. Infections were common when soft tissue was injured. From our data, we cannot conclude whether the risk for tissue infection is increased compared to any other trauma. ICAR-MEDCOM recommends that wounds be cleaned and systemic antibiotics given for open fractures [9]. Tetanus vaccinations should be up to date and are recommended, ideally prior to performing canyoning.

### 4.3. Mortality Approximation

The prospective Swiss online survey concluded that the incidence of canyoning-related injuries was 4.2/1000 hrs of canyoning [14]. Dividing the estimated total number of injured canyoneers (*n* = 1933) by this incidence results in over 460,000 hrs of canyoning in our observed timeframe (~35,000 hrs/yr). In the Swiss online survey conducted by Ernstbrunner et al., 109 participants undertook 13,690 hrs of canyoning in one season. Projecting this to our results would equal an average of 282 fulltime active canyoneers per year in the Austrian mountains. Given the nine deaths observed in our 13-year timeframe, we can also compute an absolute risk of mortality of 0.02/1000 hrs of canyoning, or 1/407 active canyoneers per year. This extrapolation is vague as the data leading to the initial 6.4 ratio were taken from only one hospital. However, given the fact that Innsbruck Medical University Hospital is a level-one trauma centre serving western Austria, some injured individuals probably also seek help from a doctor in a private practice. Therefore, the ratio of rescue to no rescue is most likely underestimated, thus overestimating the mortality rate.

A very recent study conducted by Gatterer et al. compared the mortality in different mountain sports [18]. Some of these data relied on the same national registry as our study. For the Austrian Alps, fatalities per year can therefore be summarised as approximately 110 for mountain hiking, 20 for rock climbing, 6 for via ferrata climbing and 5 for mountain biking [18,19]. With 0.7 canyoning-related deaths per year in the Austrian Alps, the absolute mortality is not far away from in the figure for being struck by lightning (0.4 deaths/year) [20]. Of course, incidences are not taken into account for these comparisons.

### 4.4. Mode of Rescue

Nearly half the canyoning-related rescues nationwide were performed by the helicopter emergency medical service (HEMS), while 27.6% were strictly ground-based. The majority of patients with trauma to the spine, pelvis, upper and lower leg, as well as shoulder and ankle, were rescued by helicopter (>70%), alone or in combination with a ground-based rescue service. Consistent with the injury severity, we found that 88.9% of patients with hand-injuries needed only ground-based rescue or no rescue at all. We suspect that patients with isolated injuries of the upper extremity were more easily able to leave the canyon on their own. Only nine patients treated at Innsbruck Medical University Hospital were transported with an organised rescue service.

The alpine regions of Austria are equipped with numerous HEMS that are easily accessible. In 2018, there were 30 HEMS bases covering the Austrian Alps, with 19 operating all year round, flying a total of more than 18,000 rescue missions [21,22]. Remoteness combined with competing private and government-funded providers can explain this high density of HEMS bases. A more frequent use of HEMS compared to regions with a lower HEMS density can therefore be expected. However, a total of 232 helicopter missions and 182 ground-based missions related to canyoning accidents in a 13-year period in Austria is not much and stands in sharp contrast to regions specialised in canyoning, where 40% of the total costs for search and rescue missions are related to canyoning [7,8].

Interestingly, the majority of patients treated at Innsbruck Medical University Hospital did not require a professional rescue mission and sought out the emergency department on their own. Merely nine patients were brought to Innsbruck Medical University Hospital by some form of emergency medical service. The national registry data showed that 314 people were involved in canyoning accidents in the state of Tyrol and 190 of those were injured. Obviously, the majority of rescued, injured patients must have been treated elsewhere. This is easily possible as there are another six smaller public and three private hospitals in Tyrol. The small number of injured patients brought to Innsbruck Medical University Hospital does, however, give a hint as to the total injury severity, as there was apparently no need for treatment at the only level-one trauma centre in western Austria.

In Austria, 24.6% of all canyoneers were uninjured but involved in rescue operations. The majority of them were either lost or trapped in a canyon or accompanying someone who was injured. With a view to the required workload for emergency medical services and the consumption of their resources, this is an interesting finding. However, the helicopter evacuation of uninjured persons can reduce the rescue time in difficult and remote terrain, thereby potentially minimising the risk of further injuries and vital risks (e.g., hypothermia).

### 4.5. Limitations

Our study has several limitations. Firstly, the data provided by the Austrian Alpine Safety Board are anonymised, so it was only possible to directly link four cases from the national registry data with in-hospital data. The accuracy of police data is unknown, though these data are often used for prosecution and insurance purposes. Additionally, it seems that, regarding our in-hospital data, many patients are treated elsewhere and a majority also come to the hospital on their own and are therefore not registered by officials in the national registry. Minor incidents which could be solved by the persons on site did not result in a call to the dispatch centre and thus were not in the national registry. The retrospective analysis of emergency medical registries is naturally biased as very minor injuries often do not require rescue operations or emergency medical services. Therefore, comparisons with prospective surveys must be conducted with caution. Additionally, comparing different retrospective analyses with differences in reasons for a rescue operation (search and rescue vs. merely rescue operations) and of course different geographic regions observed, as well as different injury definitions, limits the interpretation possibilities with regard to the entire background of observed variations in the injury severity and rate of uninjured persons.

Restrictions imposed by the Austrian Data Protection Authority prevented us from analysing hospitals other than Innsbruck Medical University Hospital. We thus do not know the incidence and severity of canyoning-related injuries or the rescue-to-no-rescue ratio elsewhere in Austria or Tyrol. Furthermore, the modality of our in-hospital analysis, namely searching for keywords, is dependent on the documentation strength. Missing cases caused by absent keywords in the documentation could result in a smaller number of hits. Moreover, the exact number of persons practicing canyoning in Austria is unknown. An exact assessment of accident incidence and mortality is therefore not possible. Finally, the rescue mode may depend on the location of the accident. Unfortunately, our databases do not provide these data.

## 5. Conclusions

Jumping, rappelling, sliding and stumbling were the most common causes of canyoning accidents. Most injuries involved the lower extremities, spine and upper extremity. Death was mainly caused by drowning, although mortality was low. A quarter of all involved persons, although uninjured, required evacuation by a professional rescue service. Increased safety precautions are required to reduce accidents while jumping and rappelling and fatalities caused by drowning.

## Figures and Tables

**Table 1 ijerph-17-00102-t001:** Characteristics of canyoning emergencies in Austria from November 1, 2005 to October 31, 2018. Activities, mode of rescue and weather are listed according to their frequency.

Characteristics		*n*	%
Sex	Female	121	25.7
	Male	348	73.9
	Unknown	2	0.4
Age	32.8 ± 10.1 yrs.		
Elevation of accident site	943 ± 328 m		
Activity before accident	Jumping	110	23.4
	Accompanying person	104	22.1
	Trapped in a canyon	57	12.1
	Rappelling	51	10.8
	Sliding	41	8.7
	Descending	33	7.0
	Stumbling	26	5.5
	Swimming	19	4.0
	Medical Emergency	6	1.3
	Falling	4	0.8
	Unknown	20	4.2
Severity of injury	Uninjured	169	35.9
	Mildly injured	96	20.4
	Severely injured	162	34.4
	Dead	9	1.9
	Undefined	35	7.4
Rescue mode	Helicopter	180	38.2
	Ground forces	130	27.6
	No rescue	71	15.1
	Ground-based rescue and helicopter	52	11.0
	Unknown	38	8.0
Weather	Sunny	279	59.2
	Cloudy	92	19.5
	Rainy	67	14.2
	Thunderstorm	10	2.1
	Fog	6	1.3
	Sudden drop in temperature	3	0.6
	Blizzard	2	0.4
	Unknown	12	2.5

Unknown: no information available. Undefined: circumstances or condition were not defined or unknown by the attending pre-hospital emergency physician. Data are presented as the mean ± SD or absolute and relative frequencies.

**Table 2 ijerph-17-00102-t002:** Activity leading to different severity grades of injuries in canyoning emergencies in Austria from November 1, 2005 to October 31, 2018.

Activity	Uninjured *n* (%)	Mildly Injured *n* (%)	Severely Injured *n* (%)	Dead *n* (%)	Undefined *n* (%)	Total *n*
Jumping	0	23 (20.9)	72 (65.5)	0	15 (13.6)	110
Accompanying person	96 (92.3)	0	0	0	8 (7.7)	104
Trapped in a canyon	53 (93.0)	4 (7.0)	0	0	0	57
Rappelling	5 (9.8)	26 (51.0)	19 (37.3)	1 (2.0)	0	51
Sliding	0	9 (22.0)	28 (68.3)	1 (2.4)	3 (7.3)	41
Descending	4 (12.1)	8 (24.2)	19 (57.6)	1 (3.0)	1 (3.0)	33
Stumbling	1 (3.8)	12 (46.2)	11 (42.3)	0	2 (7.7)	26
Unknown	10 (50.0)	5 (25.0)	4 (20.0)	0	1 (5.0)	20
Swimming	0	5 (26.3)	7 (36.8)	5 (26.3)	2 (10.5)	19
Medical emergency	0	2 (33.3)	2 (33.3)	1 (16.7)	1 (16.7)	6
Falling	0	2 (50.0)	2 (50.0)	0	0	4
Total	169 (35.9)	96 (20.4)	162 (34.4)	9 (1.9)	35 (7.4)	471 (100.0)

Unknown: no information on accident cause was available. Undefined: Severity of injury was not defined. Data are presented as absolute and relative frequencies. Percentages indicate proportions within a row.

**Table 3 ijerph-17-00102-t003:** Location of injury vs. activity in canyoning emergencies in Austria from November 1, 2005 to October 31, 2018.

Location of Injury	Rappelling *n* (%)	Descending *n* (%)	Stumbling *n* (%)	Sliding *n* (%)	Swimming *n* (%)	Jumping *n* (%)	Falling *n* (%)	Other *n* (%)	Total *n*
Head	4 (36.4)	2 (18.2)	2 (18.2)	0	1 (9.1)	2 (18.2)	0	0	11
Chest	3 (25.0)	0	0	0	4 (33.3)	1 (8.3)	0	4 (33.3)	12
Spine	4 (9.1)	0	0	2 (4.5)	0	37 (84.1)	1 (2.3)	0	44
Shoulder	7 (15.2)	4 (8.7)	5 (10.9)	5 (10.9)	4 (8.7)	18 (39.1)	0	3 (6.5)	46
Elbow	3 (37.5)	2 (25.0)	0	1 (12.5)	1 (12.5)	0	0	1 (12.5)	8
Forearm	1 (50.0)	1 (50.0)	0	0	0	0	0	0	2
Hand	5 (55.6)	2 (22.2)	2 (22.2)	0	0	0	0	0	9
Pelvic ring	1 (25.0)	1 (25.0)	1 (25.0)	0	0	1 (25.0)	0	0	4
Hip joint	1 (50.0)	0	0	1 (50.0)	0	0	0	0	2
Upper leg	1 (25.0)	0	0	0	1 (25.0)	2 (50.0)	0	0	4
Knee	1 (4.8)	4 (19.0)	6 (28.6)	5 (23.8)	1 (54.8)	3 (14.3)	1 (4.8)	0	21
Lower leg	4 (10.8)	4 (10.8)	2 (5.4)	8 (21.6)	2 (5.4)	14 (37.8)	1 (2.7)	2 (5.4)	37
Ankle joint	5 (9.4)	6 (11.3)	5 (9.4)	16 (30.2)	1 (1.9)	19 (35.8)	0	1 (1.9)	53
Foot	3 (18.8)	2 (12.5)	2 (12.5)	2 (12.5)	0	6 (37.5)	0	1 (6.3)	16
Multiple trauma	1 (33.3)	0	0	0	0	1 (33.3)	1 (33.3)	0	3
Unknown	1 (12.5)	0	0	1 (12.5)	1 (12.5)	2 (25.0)	0	3 (37.5)	8
Total	45 (16.1)	28 (10.0)	25 (8.9)	41 (14.6)	13 (4.6)	109 (38.9)	4 (1.4)	15 (5.4)	280 (100)

Unknown: no information available. Other: in all cases, none of the listed activities was performed. Data are presented as absolute and relative frequencies. Percentages indicate proportions within a row.

**Table 4 ijerph-17-00102-t004:** Location of injury vs. rescue mode in canyoning emergencies in Austria from November 1, 2005 to October 31, 2018.

Location of Injury	No Rescue *n* (%)	Ground-Based Rescue *n* (%)	Ground-Based Rescue and Helicopter *n* (%)	Helicopter *n* (%)	Unknown *n* (%)	Total *n*
Head	2 (18.2)	4 (36.4)	1 (9.1)	4 (36.4)	0	11
Chest	4 (33.3)	2 (16.7)	1 (8.3)	5 (41.7)	0	12
Spine	4 (9.1)	5 (11.4)	8 (18.2)	27 (61.4)	0	44
Shoulder	0	10 (21.7)	5 (10.9)	31 (67.4)	0	46
Elbow	2 (25.0)	3 (37.5)	0	2 (25.0)	1 (12.5)	8
Forearm	0	1 (50.0)	0	1 (50.0)	0	2
Hand	3 (33.3)	5 (55.6)	0	1 (11.1)	0	9
Pelvic ring	1 (25.0)	0	1 (25.0)	2 (50.0)	0	4
Hip joint	0	1 (50.0)	0	1 (50.0)	0	2
Upper leg	0	1 (25.0)	1 (25.0)	2 (50.0)	0	4
Knee	1 (4.8)	8 (38.1)	0	12 (57.1)	0	21
Lower leg	0	11 (29.7)	9 (24.3)	17 (45.9)	0	37
Ankle joint	2 (3.8)	13 (24.5)	5 (9.4)	33 (62.3)	0	53
Foot	4 (25.0)	5 (31.3)	1 (6.3)	6 (37.5)	0	16
Multiple trauma	0	2 (66.7)	0	1 (33.3)	0	3
Unknown	1 (12.5)	3 (37.5)	0	4 (50.0)	0	8
Total	24 (8.6)	74 (26.4)	32 (11.4)	149 (53.2)	1 (0.4)	280 (100)

Unknown: no information available. Data are presented as absolute and relative frequencies. Percentages indicate proportions within a row.

**Table 5 ijerph-17-00102-t005:** Characteristics of canyoning emergencies treated at Innsbruck Medical University Hospital from November 1, 2005 to October 31, 2018.

Characteristics		*n*	%
Sex	Female	25	43.1
	Male	33	56.9
Age	29.8 ± 10.2 yrs.		
Time of accident	13:29 ± 2:07		
Height of jump	11.3 ± 4.3 m		
Need for surgery		7	12.1
Analgesics		45	77.6
Activity before accident	Jumping	17	29.3
	Stumbling	19	32.8
	Falling	10	17.2
	Rappelling	6	10.3
	Sliding	3	5.2
	Other	3	5.2
Rescue	No rescue	48	82.8
	Ambulance	4	6.9
	Helicopter (all used rope or/winch)	4	6.9
	Ground-based emergency physician	1	1.7
	Unknown	1	1.7

Unknown: no information available. Other: in all cases, none of the listed activities were performed. Data are presented as the mean ± SD or absolute and relative frequencies.

**Table 6 ijerph-17-00102-t006:** Injured patients involved in canyoning emergencies treated at Innsbruck Medical University Hospital from November 1, 2005 to October 31, 2018.

Injury	*n* = 55	95.0%	Treatment
Mild traumatic brain injury	3	5.2	
Accidental hypothermia I	3	5.2	
Chest contusion	2	3.4	
Ischial bone fracture	1	1.7	
**Vertebral column**	**14**	**24.1**	
Single thoracic vertebral body fracture	7	12.1	Dorsal spinal fusion (*n* = 3)
Distortion of cervical column	5	8.6	
Fracture of two thoracic vertebrae	1	1.7	Dorsal spinal fusion (*n* = 1)
Fracture of two lumbar vertebrae	1	1.7	Dorsal spinal fusion (*n* = 1)
**Upper extremity**	**14**	**24.1**	
Joint or ligament of finger or hand	7	12.1	
Upper extremity	3	5.2	Debridement (*n* = 3)
Wound infection	1	1.7	
Forearm fracture	2	3.4	Osteosynthesis (*n* = 2)
Dislocated shoulder	1	1.7	
Contusion of clavicle	1	1.7	
**Lower extremity**	**25**	**43.1**	
Sprain of the ankle	6	10.3	
Sub-dislocation of the talus	5	8.6	
Lesion of the lower extremity	4	6.9	Debridement (*n* = 3)
with infection	2	3.4	
Fracture of the ankle	4	6.9	
Joint or ligament injuries, knee	3	5.2	
Distortion of adductor muscles	1	1.7	
Fracture of metatarsal bones	1	1.7	
Fracture of great toe	1	1.7	
**Illness**	***n* = 3**	**5.0%**	**Treatment**
Gastroenteritis	2	3.4	
Acute coronary syndrome	1	1.7	PTCA (*n* = 1)

PTCA denotes percutaneous transluminal coronary angioplasty; traumatic brain injury was defined as mild (loss of consciousness <30 min), moderate or severe, according to the American Congress of Rehabilitation Medicine (ACRM) definition [11,12]. Accidental hypothermia was graded in accordance with the Swiss Staging of Hypothermia, where accidental hypothermia I is defined as a body temperature of 32 to 35 °C [13]. Injuries per body region are highlighted in bold. Lesions were any injuries to the tissue not defined as any other injury. Data are presented as absolute and relative frequencies.

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
