# Peer review of "Canyoning Accidents in Austria from 2005 to 2018"

_ijerph, 2019, doi:10.3390/ijerph17010102_

Round 1

Reviewer 1 Report

see attachment

Reviewer 2 Report

Dear authors,

thanks a lot for providing this paper being reviewed.
In general, it is an interesting topic and related literature is scarce. Therefore, analysing this growing wilderness receational sport regarding accidenct causes, injury patterns etc. is the next logical step to further decrease the injury risk within this outdoor sport.
I recommend this paper potentially acceptable for publication, once some essentiel revisions have been carried out. To enable a point to point revision, proposed corrections are referred to the line numbers.

#35: Are there any studies reporting the amount of people enjoying canyoning (Austria, Europe, worldwide etc.)?

#45/49: Did the studies [7,8] evaluate the injury risk (injuries/1000 hours etc.)? 

#63: was registered

#63: better use the term "members of the Federal Ministry of the Interior, who are qualified alpinists and have paramedical training..." instead of "Alpine Police officers..."

#67: ...registries for mountain accidents in Austria

#73/74: I would recommend to list all potential categories of mentioned data; Accidentc cause (jumping, accompanying Person, trapped...etc.)

#75: of all victims

#94: According to Figure S2 - is the number of n=471 canyoning accidents correct because after excluding all data as presented in your flowchart, n=483 would remain; please clarify!

#94:, or absolute and relative frequencies

#103: 471? see comment to line # 94

#108: Table 1 reports data on sex, Age, Elevation of accident side, activity before accident, severity of injury, rescue mode and weather.

#111: injuries were suffered by n=284 but the alpine safety board showed 297 accidents - how is the number n=284 calculated?

#113-116: To which cause of accident do the mentioned numbers in lines 113-116belong to?

#113: rappeling also accounts for 37.3% of severe injuries, add this activity to descending.

#125:Injuries to the lower extremity, shoulder and to the spine…

#133: Use the same terms for the rescue mode in all tables (Table 1 compared to Table 4 – Ground forces vs. terrestrial rescue)

#135: The notes of the tables should not contain information that is already presented in the method section; Generally I would recommend: “Data are presented as absolute and relative frequencies…etc”.

#139: The notes of the tables should not contain information that is already presented in the method section; Generally I would recommend: “Data are presented as absolute and relative frequencies…etc”.

#144: The notes of the tables should not contain information that is already presented in the method section; Generally I would recommend: “Data are presented as absolute and relative frequencies…etc”.

#144: Within the results section, it is mentioned that injuries were suffered by n=284 persons. Why does table 2 present a total number of injuries of n=280

#148: Generally I would recommend: “Data are presented as absolute and relative frequencies…etc”.

#148: Within the results section, it is mentioned that injuries were suffered by n=284 persons. Why does table 2 present a total number of injuries of n=280

#153: Males were more frequently involved – please state adequate numbers of both sexes (%) – if so, please add this information to the results section of the national registry data from the Austrian Alpine Safety Board – line #102

#171/175: The notes of the tables should not contain information that is already presented in the method section; Generally I would recommend: “Data are presented as absolute and relative frequencies…etc”.

#186: Jumping, rapelling, sliding and stumbling were the most common… - is this result presented in the results section?

#187: replace upper extremity by shoulder

#188: Number in parenthesis

#188: a quarter? – to which number do you refer? Where is this number already mentioned? According to table 2, 35.9% of all analyzed cases were uninjured – please clarify!

#194:state numbers regarding sex ratios of other registries

#211-213: These sentence is not quite clear, please clarify

#218: Where does this result come from? I am not able to find this result neither within the results section nor in any of the tables.

#219: from Switzerland

#240: see comment to line#188

kindest regards!

Reviewer 3 Report

This is an interesting study and the authors have collected a wide enough dataset using a correct methodology. The paper is generally well written and structured. They have reviewed the most relevant previous studies and analysis was well performed.

Author Response

Dear Reviewer,

Thank you very much for your review.

We are confident that there are no concerns regarding our manuscript.

Kind regards.